# Curcumin: A Novel Way to Improve Quality of Life for Colorectal Cancer Patients?

**DOI:** 10.3390/ijms232214058

**Published:** 2022-11-14

**Authors:** Laura Layos, Eva Martínez-Balibrea, Vicenç Ruiz de Porras

**Affiliations:** 1Medical Oncology Department, Catalan Institute of Oncology, Ctra. Can Ruti-Camí de les Escoles s/n, 08916 Badalona, Spain; 2Catalan Institute of Oncology, Badalona Applied Research Group in Oncology (B·ARGO), Ctra. Can Ruti-Camí de les Escoles s/n, 08916 Badalona, Spain; 3Germans Trias i Pujol Research Institute (IGTP), Ctra. Can Ruti-Camí de les escoles s/n, 08916 Badalona, Spain; 4ProCURE Program, Catalan Institute of Oncology, Ctra. Can Ruti-Camí de les escoles s/n, 08916 Badalona, Spain

**Keywords:** curcumin, metastatic colorectal cancer, chemotherapy, chemotherapy-related toxicity, chemoresistance, quality of life

## Abstract

Colorectal cancer (CRC) is the third most common cancer in men and the second most common in women. Treatment of metastatic CRC consists of highly toxic chemotherapeutic drug combinations that often negatively affect patient quality of life (QoL). Moreover, chemotherapy-induced toxicity and chemotherapy resistance are among the most important factors limiting cancer treatment and can lead to the interruption or discontinuation of potentially effective therapy. Several preclinical studies have demonstrated that curcumin acts through multiple cellular pathways and possesses both anti-cancer properties against CRC and the capacity to mitigate chemotherapy-related side effects and overcome drug resistance. In this review article, we suggest that the addition of curcumin to the standard chemotherapeutic treatment for metastatic CRC could reduce associated side-effects and overcome chemotherapy resistance, thereby improving patient QoL.

## 1. First-Line Treatment of Metastatic Colorectal Cancer: An Overview

Colorectal cancer (CRC) is the third most common tumor and the second leading cause of cancer death worldwide [1,2]. Metastases, the greatest cause of cancer-related mortality, are present in nearly 25% of newly diagnosed CRC patients, up to 40% of whom will relapse during follow-up after curative primary tumor surgery. Over the last ten years, new combinations of cytotoxic agents and targeted therapies have improved the prognosis of metastatic CRC (mCRC), and median overall survival is now 30 months, highlighting the importance of a “continuum of care” approach in advanced disease [3]. Approximately one-third of mCRC patients have limited-liver metastatic disease and could be candidates for surgery with curative intent after systemic treatment [4].

The backbone of first-line treatment of mCRC consists of doublets or triplets of fluoropyrimidines (thymidylate synthase (TS) inhibitors) such as 5-fluorouracil or capecitabine in combination with oxaliplatin (a platinum DNA damage agent) and/or irinotecan (a topoisomerase I inhibitor). The addition of targeted therapies such as anti-epidermal growth factor receptor (EGFR) monoclonal antibodies (cetuximab or panitumumab) or antiangiogenic therapy (bevacizumab or aflibercept) in first or later lines improves both overall survival and response rates. The clinical benefit of EGFR inhibitors is limited to RAS-wild-type tumors (KRAS exons 3-4 and NRAS exons 2-3-4), which are present in nearly 50% of newly diagnosed patients, and the response seems to be superior in left-side primary tumors [5,6,7,8,9,10]. In addition, 4–5% of mCRC patients harbor BRAF mutations (most frequently V600E), in which the combination of specific inhibitors, such as encorafenib, with anti-EGFR therapy has been shown to increase overall survival [11]. Other targeted therapies such as regorafenib (a multitargeted kinase-inhibitor) and ramucirumab (angiogenesis inhibitor) have also shown good results in chemorefractory mCRC [12,13]. In addition to agents blocking pathways involved in tumor growth and spread, accumulating evidence has shown that targeting pathways involved in immunomodulation is also effective. Immune checkpoint inhibitors stop the tumor from escaping T cell detection and thus stimulate immune surveillance and clearance [14,15]. However, the efficacy of immune checkpoint inhibitors is basically limited to tumors with deficient Mismatch Repair (dMMR) and a high mutational burden, which comprise only 4–8% of all mCRC cases [16,17]. The programmed death-ligand 1 (PD-1) blocker pembrolizumab has indeed shown promising results as a monotherapy in this subgroup of patients. The KEYNOTE-177 study comparing pembrolizumab with chemotherapy in dMMR patients reported a progression-free survival of 16.5 months with pembrolizumab compared to 8.2 months for the chemotherapy group. Moreover, the duration of response at 24 months was 83% compared to 35%, respectively [18]. Equally encouraging results have been achieved with another PD-1 inhibitor nivolumab and the cytotoxic T-lymphocyte antigen 4 (CTLA-4) inhibitor ipilimumab [19].

## 2. Chemotherapy-Associated Toxicity in mCRC

The most common first-line treatment regimens in unresectable mCRC patients are FOLFOX (folinic acid + 5-fluorouracil + oxaliplatin) and FOLFIRI (folinic acid + 5-fluorouracil + irinotecan) in combination with either an anti-EGFR antibody (in RAS-wild-type tumors) or with bevacizumab. Unfortunately, all these combinations are extremely toxic and can compromise patient quality of life (QoL), which is increasingly being recognized as both a crucial outcome in clinical practice and an endpoint in randomized clinical trials [20,21]. In fact, chemotherapy-induced toxicity is one of the most influential factors limiting cancer treatment and is often associated with the interruption or even discontinuation of potentially effective anti-cancer therapy.

Approximately 30% of patients treated with fluoropyrimidines develop severe toxicities (≥Grade III Common Terminology Criteria for Adverse Events), including myelosuppression, severe diarrhea, vomiting, stomatitis (inflammation of the mucus lining in the mouth), mucositis, hand-foot syndrome (palmar-plantar erythrodysesthesia), and neuropathy. Moreover, fluoropyrimidine-related toxicity leads to death in 0.5–1% of patients [22,23,24,25]. Oxaliplatin also has severe side effects, such as gastrointestinal upset (nausea and vomiting, diarrhea, and mucositis), hematological disorders (anemia, thrombocytopenia, and neutropenia), peripheral neuropathy, and hepatotoxicity [26]. In fact, neuropathy, the major problem associated with oxaliplatin treatment, occurs in up to 70% of patients and leads to dose limitation and treatment discontinuation [27,28]. It has been suggested that the development of neurotoxicity, hepatotoxicity, and nephrotoxicity is at least partly due to oxaliplatin-induced oxidative damage to mitochondria and to the inhibition of sodium pumps by the chelating action of oxalate on calcium and magnesium molecules [29,30]. Management of these adverse effects is based on the administration of calcium gluconate and/or magnesium sulphate [31]. Irinotecan is also often associated with severe toxicities, especially neutropenia and diarrhea, generally caused by the insufficient glucuronidation of the irinotecan active metabolite SN-38 by the UDP-glucuronosyltransferase (UGT) 1A1 (UGT1A) enzyme. The resulting elevated SN-38 plasma concentration is responsible for the often life-threatening hematological and gastrointestinal toxicities associated with irinotecan [32]. Finally, targeted therapies such as bevacizumab have been associated with thromboembolic events and the occurrence of grade 3/4 hypertension and bleeding in 2% to 4% of patients [33]. Fortunately, the addition of bevacizumab to FOLFOX seems to be well tolerated and does not markedly change the overall chemotherapy-related toxicity [23].

## 3. Curcumin Attenuates Chemotherapy-Related Toxicity

For many years, curcumin (diferuloylmethane)—the “golden spice”—has been widely studied because of its pleiotropic effects in cancer. Curcumin, a hydrophobic polyphenol, is derived from the rhizome of the herb *Curcuma longa* and constitutes the major curcuminoid in the spice turmeric (77% curcumin, 17% demethoxycurcumin, 3% bis-demethoxycurcumin). Curcumin is “generally recognized as safe” (GRAS) as a dietary supplement by the U.S Food and Drug Administration (FDA) and the European Food Safety Authority (EFSA) and has been catalogued with the E100 code of the European Union. One of the clinical benefits of curcumin is the improvement of QoL in several health conditions [34], including cancer [35,36].

Curcumin is a pleiotropic agent that acts through multiple cellular pathways and has been shown to possess anti-cancer properties against CRC in vitro and in vivo [37,38]. Many of its anti-cancer properties have been attributed to its role as an anti-inflammatory and antioxidant, as well as to its ability to modulate the cell cycle and the pathways involved in proliferation, apoptosis, migration, invasion, angiogenesis, and metastasis [39], which are typically targeted by the drugs used to treat CRC. Mechanistically, curcumin modulates several CRC molecular targets at the same time—either by altering their gene expression, activation, or signaling pathways, or by direct interaction [37,38,39]. Importantly, in addition to its well-known anti-cancer properties, curcumin can also alleviate some of the chemotherapy-related side effects [40]. For example, curcumin attenuates the liver injury induced by oxaliplatin through activation of the nuclear factor-erythroid 2-related factor 2 (Nrf2) signaling, a key regulator pathway of cellular defense against oxidative and electrophilic stresses [41], as well as the nerve damage and the oxidative damage to mitochondria caused by oxaliplatin [42]. In fact, curcumin has been shown to not only hinder mitochondrial damage but also to protect mitochondria and induce activity of mitochondrial complex enzymes [36,42,43]. Interestingly, similar effects of curcumin on cisplatin-related toxicity have been observed in several tumor types [44,45,46,47,48]. Additionally, curcumin protects against irinotecan-induced intestinal injury by inhibiting nuclear factor kappa B (NF-κB) transcription factor activation [49], and it is also active against FOLFIRI-related cardiovascular toxicity [50] and capecitabine-induced hand-foot syndrome [51]. Recently, it has been shown that curcumin attenuates bevacizumab-associated cardiotoxicity by suppressing oxidative stress and preventing mitochondrial dysfunction in heart mitochondria [52].

In a study of curcumin’s effects in cancer patients, Belcaro and colleagues looked at the side effects of chemotherapy in several tumor types, including colon, ovarian, lung, liver, kidney, and stomach cancers. Of 80 patients treated with chemotherapy, 40 simultaneously received 500 mg of curcumin. Chemotherapy-related nausea, diarrhea, constipation, weight loss, neutropenia, and cardiotoxicity were significantly lower in the patients receiving curcumin than in the control group. Moreover, patients receiving curcumin also required fewer medications for treating these side effects [53]. In the same vein, turmeric supplementation for 21 days resulted in a clinically relevant and statistically significant improvement in global health status, symptom scores (fatigue, nausea, vomiting, pain, appetite loss, insomnia), and hematological parameters of breast cancer patients treated with paclitaxel [54]. Taken together, these findings lead us to suggest that the addition of curcumin to the standard treatment of CRC could not only attenuate chemotherapy-associated side effects but also improve the QoL of patients (Figure 1).

## 4. Curcumin Reverts Chemotherapy Resistance in mCRC

In addition to chemotherapy-related toxicity, chemoresistance remains one of the main problems hindering treatment success. Tumor cells can be intrinsically resistant or acquire resistance during a treatment. Resistance to chemotherapy is a complex and multifactorial process involving several mechanisms, including drug influx/efflux modifications, alterations in DNA damage repair (DDR), decreased cell death activation, autocrine survival signaling, and high detoxification activity [55,56]. One of these mechanisms with consequences in mCRC is the hyperactivation of the NF-κB signaling pathway [57], which promotes the expression of several target genes involved in inflammation, cell proliferation, apoptosis, angiogenesis, invasion, metastasis, and chemoresistance [58,59]. In fact, most of the anti-inflammatory and anti-cancer properties of curcumin are believed to be due to its ability to inhibit NF-κB activation through interaction with the IκB kinase complex (IKK) by inhibiting the phosphorylation and degradation of IκBα, a NF-κB inhibitor, and thereby blocking the nuclear translocation of this transcription factor [37,60,61]. Along with other studies [62,63,64], our group has demonstrated that curcumin can overcome oxaliplatin resistance by inhibiting the activity of the CXC-chemokines/NF-κB axis and, consequently, the expression of genes involved in anti-apoptosis and proliferation [57]. Additionally, in CRC preclinical models, curcumin was shown to enhance the effect of 5-fluorouracil [65,66] and capecitabine [67] by inhibiting AKT and NF-κB activity, and consequently, NF-κB-regulated gene products. In the same vein, Pattel and colleagues reported that curcumin sensitizes CRC cells to FOLFOX by inhibiting EGFR family receptors and insulin-like growth factor-1 receptor (IGF-1R) [68,69,70], the overexpression of which has been related to chemoresistance in CRC [71,72].

Chemotherapy resistance is also related to the specific mechanism of action of the drug. An example of such a specific mechanism is gene amplification in TS in 5-fluorouracil treated patients [73] and upregulation of genes involved in DDR pathways, such as ERCC1 in oxaliplatin treated patients [74]. Interestingly, Rajitha and colleagues demonstrated that the inhibition of NF-κB translocation by curcumin or its analogs induces cell cycle arrest and downregulates TS in CRC cells [61]. Furthermore, curcumin was found to inhibit ERCC1 through its ability to modulate miR-409-3p, thereby overcoming oxaliplatin resistance in CRC cells [75].

Curcumin can also promote the activation of apoptotic pathways by increasing the generation of reactive oxygen species (ROS) [76]. In a recent work, Li and colleagues demonstrated that curcumin can reverse Nicotinamide *N*-methyltransferase-induced cell proliferation and 5-fluorouracil resistance through ROS generation and cell cycle arrest [77].

On the other hand, the drug-resistant phenotype is associated with the acquisition of mesenchymal features, and epithelial-to-mesenchymal transition (EMT) plays a key role in chemoresistance in CRC, mainly through the activation of the NF-κB and transforming growth factor β (TGF-β) pathways [78,79,80]. In fact, EMT was observed in chemotherapy-resistant CRC cell lines [57,81,82], while curcumin was able to revert this chemoresistance by downregulating EMT markers [83] through TGF-β/Smad2/3 signaling attenuation [84], by upregulating EMT suppressive miRNAs [85] or by downregulating the TET1-NKD2-WNT signaling pathway [86]. In addition, several studies have demonstrated that curcumin can sensitize colon cancer stem cells (CSC), a small subpopulation of cells within tumors capable of self-renewal, differentiation, and tumorigenicity [87], to 5-fluorouracil, FOLFOX and irinotecan, thereby preventing the emergence of chemoresistant CRC cells [70,88,89,90,91]. In this regard, a recent study has demonstrated that treatment of CRC organoids with a combination of amorphous curcumin (a compound with improved solubility and bioavailability) and oxaliplatin, 5-fluoroouracil, or irinotecan showed a synergistic activity through the inhibition of proliferation-related signals and CSC marker expression, in addition to arresting the ERK signaling pathway [92]. Along the same lines, Zheng and colleagues showed that low doses of curcumin promoted the sensitivity of CRC cells to 5-fluorouracil by downregulating phospho-ERK signaling [93].

Finally, several studies have shown that curcumin can increase the intracellular accumulation of oxaliplatin and 5-fluorouracil in CRC cells by downregulating the P-gp [75,94] and ATP-binding cassette transporter G2 (ABCG2) [70] drug-efflux transporters both at the mRNA and protein levels. Preclinical data have suggested that the expression of ATP-binding cassette (ABC) transporters, such as ABCC2 [95], ABCB4 [96], as well as the multidrug resistance protein 1 (MDR1, also known as P-glycoprotein or P-gp), which is encoded by ABCB1 [97,98], can confer resistance to chemotherapy. However, evidence that these transporters contribute to drug resistance in human tumors is sorely lacking [99] and the development of MDR1 as a therapeutic target has been unsuccessful [100]. It is important to highlight that although several studies have related the ABC transporters’ overexpression to platinum resistance [55,101,102], the association between oxaliplatin resistance and the MDR1 expression has shown unconvincing results. For instance, Ekblad and colleagues described an overexpression of this membrane transporter as a consequence of oxaliplatin resistance acquisition in vitro, although functional tests did not show any increase in ABCB1 transport activity in the oxaliplatin-resistant models compared with its parental cell lines [103]. Other studies have reported no association between these drug efflux pumps and the sensitivity to oxaliplatin in CRC clinical samples [104]. In the same vein, the ability of MDR1 to confer resistance to 5-fluorouracil and irinotecan has been demonstrated in different CRC cell lines transfected with this carrier. However, its clinical relevance in CRC refractoriness to antitumor chemotherapy remains to be established [105,106]. Taken together, these results highlight the necessity of further investigation into the role of MDR1 and curcumin in oxaliplatin and 5-fluorouracil resistance in CRC patients.

Most clinical data on curcumin come from early phase clinical trials, with results showing that oral curcumin can achieve efficacious levels in the colon with negligible distribution outside the gut [107,108]. Moreover, curcumin was shown to be safe in advanced CRC patients when administered for up to four months [109]. In addition, a study by James and colleagues found that curcumin at doses up to 2 gms daily was highly tolerable when added to a FOLFOX regimen in mCRC patients with liver metastases [110]. More recently, the same group performed a phase IIa randomized trial of first-line treatment for mCRC patients comparing FOLFOX +/−bevacizumab with the same regimen plus curcumin 2 gms/day (CUFOX) in mCRC patients. In the intention-to-treat population, patients in the CUFOX arm achieved longer overall survival (HR 0.34; *p* = 0.02) but there was no difference in progression-free survival (HR 0.57) [111].

In conclusion, a further improvement in outcomes for mCRC highly depends on identifying and targeting mechanisms of drug resistance. Taken together, these findings offer compelling evidence that combining curcumin with conventional chemotherapy may be effective in overcoming drug resistance in mCRC (Table 1).

## 5. Conclusions

Treatment for mCRC consists of highly toxic drug combinations that often negatively affect the QoL of patients. We believe that patient QoL must be recognized as an essential outcome in clinical practice; moreover, it is increasingly being reported as an endpoint in randomized clinical trials. Optimizing strategies to control chemotherapy-related toxicity will not only improve patient QoL but will also improve adherence to cancer treatment and thus improve patient survival. This is especially crucial for mCRC patients, in whom chemotherapy is prescribed as a neoadjuvant treatment before surgery for liver metastases, where it is critical to maintain an adequate dose intensity in order to proceed to curative surgical treatment. Furthermore, our own experience has taught us that the unprescribed use of several plant-derived supplements is very common among cancer patients even when their positive effect on QoL has not been demonstrated in clinical studies. One of the most commonly used herbal supplements is curcumin, which has been extensively studied in cancer prevention and treatment. In fact, a plethora of preclinical studies have demonstrated the anti-cancer properties of curcumin as well as its role as a chemosensitizer agent [37,38,39]. Several preclinical studies have demonstrated that the addition of curcumin to the standard treatment of CRC could decrease treatment-associated side effects and enhance chemotherapy efficacy [40,53]. Therefore, considering that therapy-induced toxicity is among the most important factors limiting cancer treatment and is usually associated with discontinuation of potentially effective therapy, we suggest that adding curcumin, a natural compound with a very low toxicity profile in humans [39], to current mCRC treatment regimens could be a potential synergistic strategy to reduce chemotherapy-related adverse effects, improve treatment efficacy, and decrease drug resistance.

Additionally, it is important to identify predictive biomarkers of response to curcumin-based treatment. To the best of our knowledge, only a few studies have focused on this question. However, in a previous study by our group, we found that treatment with oxaliplatin induces the expression of the CXCL1 chemokine that was repressed by the addition of curcumin—both in CRC cell lines and in patient-derived CRC liver metastasis explants treated with oxaliplatin or oxaliplatin + curcumin. Interestingly, the explants with the “best response” to oxaliplatin + curcumin were those with the highest baseline levels of CXCL1, suggesting that this chemokine could be a good predictive marker for this treatment [57]. Prompted by these observations, Howells’ group conducted a phase IIa trial in which they assessed CXCL1 plasma levels in patients receiving FOLFOX or CUFOX. Although there was no significant difference in plasma CXCL1 concentrations after curcumin treatment, mean baseline concentrations were 1.7-fold higher in FOLFOX patients than in CUFOX patients [111]. In the same vein, Lu and collaborators recently demonstrated that CRC patients with microsatellite-stable tumors and high baseline IκBα protein expression would benefit from curcumin treatment [112].

Finally, improved delivery strategies and new curcumin formulations (such as nanoparticles, liposomes, and synthetic analogues) will increase the absorption and bioavailability of curcumin [113].

Certainly, further research is warranted on the potential role of curcumin in reducing chemotherapy-induced toxicity and on predictive biomarkers to identify those patients most likely to benefit. Unfortunately, the few clinical trials of curcumin performed to date have often been limited by wide patient heterogeneity, small sample size, and the poor bioavailability of the curcumin formulations studied. For this reason, we strongly recommend that randomized, double-blind, placebo-controlled trials of bioavailable curcumin be carried out. The results of such trials will further elucidate the role of this polyphenol in overcoming chemoresistance and improving the QoL of mCRC patients.

## Figures and Tables

**Figure 1 ijms-23-14058-f001:**
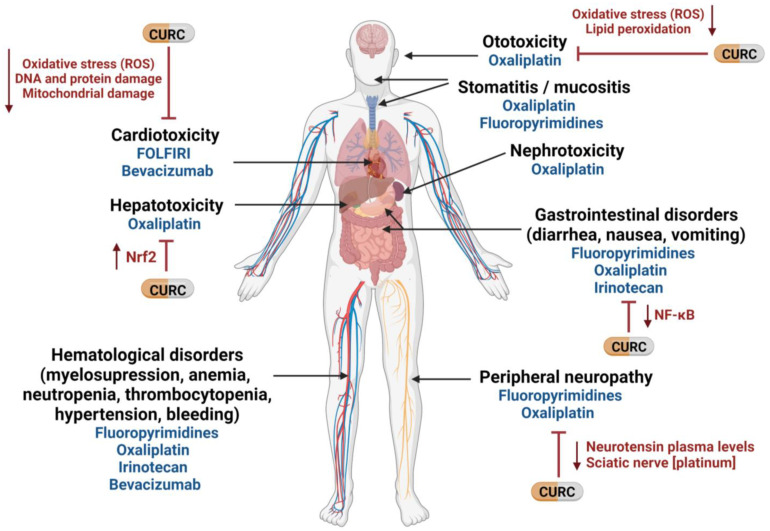
Potential use of curcumin to mitigate therapy-related side effects in mCRC. CURC: curcumin; NF-κB: nuclear factor kappa B; Nrf2: nuclear factor -erythroid 2- related factor 2; ROS: reactive oxygen species. Created with BioRender.com (accessed on 16 September 2022).

**Table 1 ijms-23-14058-t001:** Main molecular mechanisms of action of the combination of curcumin and chemotherapeutic agents in preclinical models of CRC. ABCG2: ATP-binding cassette transporter G2; CSCs: cancer stem cells; EGFR: epidermal growth factor receptor; EMT: epithelial-to-mesenchymal transition; IGF-1R: growth factor-1 receptor; NF-κB: nuclear factor kappa B; P-gp: P-glycoprotein; ROS: reactive oxygen species; TGF-β: transforming growth factor β; TS: thymidylate synthase.

Treatment Regimen	Molecular Targets of Curcumin	References
Oxaliplatin + Curcumin	Inhibition of NF-κB activation	[57,62,63,64]
	Downregulation of CXCL8, CXCL1 and CXCL2 chemokines	[57]
	Inhibition of AKT activation	[57]
	Inhibition of miR-409-3p/ERCC1 axis	[75]
	TGF-β/SMAD2/3 signaling attenuation	[84]
	P-gp downregulation	[75,94]
5-fluorouracil + Curcumin	Inhibition of NF-κB activation	[61,65,66]
	Downregulation of TS	[61]
	P-gp downregulation	[75,94]
	ROS generation	[77]
	Downregulation of TET1-NK2-WNT pathway	[86]
	Elimination of CSCs	[89]
	Upregulation of EMT-suppressive miRNAs (miR-200b, miR-200c, miR-141, miR-429, miR-101t)	[85]
	Inhibition of ERK signaling pathway	[92,93]
FOLFOX + Curcumin	Downregulation of EGFR and IGF-1R	[68,69,70]
	Inhibition of NF-κB activation	[70]
	Inactivation of β-catenin, COX-2, c-Myc and Bcl-xL	[70]
	Elimination of CSCs	[70,88]
	Downregulation of miR-21	[91]
	Downregulation of ABCG2 drug-efflux transporter	[70]
Irinotecan + Curcumin	Downregulation of EMT markers (Vimentin and N-cadherin)	[83]
	Elimination of CSCs by apoptosis induction	[90]

## Data Availability

Not applicable.

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
