# Peer review of "Curcumin: A Novel Way to Improve Quality of Life for Colorectal Cancer Patients?"

_ijms, 2022, doi:10.3390/ijms232214058_

Round 1

Reviewer 1 Report

This article does not contain in-depth information to publish in this reputed journal.

Reviewer 2 Report

There are two major issues which i feel must be fully explained by the author team before any decision should be taken by the publishing team.

I find that there is very little differentiation provided by the author between this review and their previous review which I accept is referenced in this manuscript and there is no attempt by the team to obfuscate this.

De Porras, V. R., Layos, L., & Martínez-Balibrea, E. (2020). Curcumin: A therapeutic strategy for colorectal cancer? Seminars in Cancer Biology. doi:10.1016/j.semcancer.2020.09.004

The second aspect is that this is a very active  area of research and there is a lack of recent publications on the topic  cited in the submitted manuscript. In fact only 4 papers dated from 2022 are cited some of which themselves are review articles.

If one looks at Pubmed/Sciencedirect there are close to 500 articles in the period 2021-2023. It is conceivable that the authors are looking to use the information collected from the substantial literature review carried out for the original article for a new publication.

This review could be acceptable if more recent references were included.

Reviewer 3 Report

The manuscript entitled as “Curcumin for reduction of chemotherapy-associated toxicity and chemo-2 resistance: a novel way to improve quality of life for metastatic colorectal cancer patients?” aimed to describe the possibility of reduction of chemotherapy-associated toxicity and chemo-resistance by curcumin for metastatic colorectal cancer patients. This is a clinical problem that reflects the cytotoxicity of chemotherapy and chemo-resistance in the treatment of colorectal cancer and cancer metastasis. Addition of Curcumin as a chemo-adjuvant might have the potential to improve quality of life for metastatic colorectal cancer patients, as authors showed in this manuscript. 

The overall impression of this review is that the authors picked an important area to review. A major improvement is needed to be accepted for publication.

1.      Must cite original research (enough) to justify Curcumin could potentially aid in overcoming the cytotoxicity of chemotherapy and chemo-resistance.

2.      The manuscript is lack of cohesion. There are a couple of disorganizations. For example, Page no. 5, 2nd last paragraph (Finally, a large body of preclinical…) has no connection with Curcumin. A lot of typos (Metastases line 32) and grammatical error make this review difficult to read.

3.      The title should summarize the main idea or ideas of the study. A decent title comprises the least possible words that effectively describe the contents and/or purpose of your review article. For example, the title contains “metastatic colorectal” but authors did not provide a single citation that curcumin positively impact metastatic colorectal cancer patients.  

4.      Although the authors try to provide a future perspective, however, ended up with writing the review again.

Round 2

Reviewer 1 Report

The article is now improved. 

Reviewer 2 Report

I am happy that the authors have taken the time to add new references from more recent publications. This action will ensure that some of the data presented will not be out of date by the time of publication.

The amended title better reflects the content of the review and crucially differentiates it from the authors previous review.

Some small editing and proofreading required

Reviewer 3 Report

Much improved, however, moderate English changes required to enhance readability.